# YOLO-RFF: An Industrial Defect Detection Method Based on Expanded Field of Feeling and Feature Fusion

**Gang Li** \*, **Shilong Zhao**, **Mingle Zhou**, **Min Li**, **Rui Shao**, **Zekai Zhang** **and Delong Han** \*

Shandong Computer Science Center (National Supercomputer Center in Jinan), Qilu University of Technology
(Shandong Academy of Sciences), Jinan 250353, China
\* Correspondence: lig@qlu.edu.cn (G.L.); handl@qlu.edu.cn (D.H.)

**Abstract:** Aiming at the problems of low efficiency, high false detection rate, and poor real-time performance of current industrial defect detection methods, this paper proposes an industrial defect detection method based on an expanded perceptual field and feature fusion for practical industrial applications. First, to improve the real-time performance of the network, the original network structure is enhanced by using depth-separable convolution to reduce the computation while ensuring the detection accuracy, and the critical information extraction from the feature map is enhanced by using MECA (More Efficient Channel Attention) attention to the detection network. To reduce the loss of small target detail information caused by the pooling operation, the ASPF (Atrous Spatial Pyramid Fast) module is constructed using dilate convolution with different void rates to extract more contextual information. Secondly, a new feature fusion method is proposed to fuse more detailed information by introducing a shallower feature map and using a dense multiscale weighting method to improve detection accuracy. Finally, in the model optimization process, the K-means++ algorithm is used to reconstruct the prediction frame to speed up the model's convergence and verify the effectiveness of the combination of the Mish activation function and the SIoU loss function. The NEU-DET steel dataset and PCB dataset is used to test the effectiveness of the proposed model, and compared to the original YOLOv5s, our method in terms of mAP metrics by 6.5% and 1.4%, and in F1 by 5.74% and 1.33%, enabling fast detection of industrial surface defects to meet the needs of real industry.

**Keywords:** industrial defect detection; perceptual field; dense multiscale weighted feature fusion; MECA attention; ASPF module





## 1. Introduction

The detection of surface defects in industrial products is an important part of industrial production and helps to safeguard the quality of products. In the process of manufacturing industrial products, due to the production environment, equipment and other aspects, a variety of defects can arise during the production process, which can shorten the life of the product and affect its quality. However, these problems can be avoided to a certain extent if the material is checked for defects prior to processing and production. Therefore, it is essential to investigate an industrial inspection algorithm to improve the accuracy and speed of defect detection. Currently, industrial research based on deep learning can fully exploit the potential features of the data without the need to manually design features, with high accuracy and robustness, and has been applied to various research areas. For example, Li et al. [1] proposed an attention-based augmentation approach to address defect detection on the surface of industrial products by using an attention enhancement method that makes the model focus more on the places that need attention; Zhang et al. [2] used an attention and knowledge distillation approach for industrial defect detection, using a large model to guide a small model to detect different classes of defects, and adding an attention mechanism to drive the model to focus on the locations of defects; Mei et al. [3]

used a hierarchical idea of image pyramid combined with convolutional denoising coding network (CDAE) to achieve defect detection on image surfaces; Guo et al. [4] proposed a Transformer-based method for steel surface defect detection, using the Transformer to provide more layers of features in combination with global contextual information to improve the recall of defective targets; Luo et al. [5] proposed a surface defect detection algorithm based on YOLO feature enhancement, which improved the detection speed to a certain extent, but the detection accuracy was low; Liu et al. [6] proposed cascaded YOLO model-based aerial image insulator identification and defect detection, using YOLOv3-dense model for defect localization and YOLOv4-tiny model for defect identification, and finally cascaded the two models to solve the problem of insulator defect detection speed and accuracy. Therefore, our task was to design a real-time, end-to-end industrial inspection method. The one-stage detector YOLO [7] can achieve the expected real-time accurate effect, but due to the particularity of some industrial image production, the foreground and background difference of the image are not obvious, and the defect target is smaller than the original image, which makes the network unable to learn the defect characteristics well. To address the above problems, firstly, to improve the detection speed of the model, we designed a new feature extraction network using deep separable convolution and MECA attention module to reduce the model computation; secondly, we proposed a new dense multiscale weighting strategy in the neck network to improve the sensitivity to small target detection by using the features extracted from more shallow layer networks; finally, to select the best last frame, we used a K-means++ clustering algorithm to filter the previous frame. The effectiveness of the innovative points in this paper was proved in NEU-DET steel dataset and PCB dataset. To summarize, the main contributions of this paper are as follows:

(1) A new feature extraction network is proposed to improve the original convolution module by using depth-separable convolution to reduce the computational effort based on its feature extraction capability.

(2) To prevent the pooling operation from causing the loss of more detailed information when extracting features, the pooling operation in the fast spatial pyramid (Spatial Pyramid Pooling—Fast, SPPF) module is replaced by the dilate convolution, which makes the extracted feature map contains more contextual information, increases the perceptual field while avoiding the loss of small target information, and introduces the MECA channel attention proposed in this paper to enhance the attention to important channels and improve the model's extraction of multiscale detail information

(3) A new dense multiscale weighting strategy is proposed to introduce a shallower layer of feature map to increase the detail information of the image, and enhance the spatial position between adjacent feature layers information to improve the detection accuracy of small targets.

(4) In the process of model optimization, the K-means++ clustering algorithm is used to reconstruct the prediction frame to avoid the problem of positive and negative sample imbalance and make it more suitable for small target datasets to improve detection accuracy.

(5) The effectiveness of the combination of the Mish activation function and SIoU loss function is verified through experiments.

The rest of the paper is structured as follows: Section 2 introduces the related work, Section 3 describes the methods highlighted in this paper in detail, Section 4 pin discusses the experimental results on the steel dataset and PCB dataset, and Section 5 discusses the experimental results as a whole. Section 6 summarizes the conclusions and future prospects of this paper.

## 2. Related Work

### 2.1. Feeling the Wild

Expanding the perceptual field is an essential tool to improve target detection performance; before the emergence of dilate convolution [8], the area was usually increased by

downsampling and extensive use of convolutional layers. As the network model deepens, however, the field is expanded, and more semantic feature information is extracted; the pooling operation will cause a reduction in image resolution, resulting in the loss of small target information in the detection process, while the model The number of parameters increases and the training speed slows down. In contrast, the dilate convolution can expand the field of perception with little change in the number of parameters and extract more image detail information.

### 2.2. Attention Mechanism

The attention mechanism [9] can assign weights according to the importance of the target and highlight certain important features to effectively capture contextual information, which has achieved good results in several computer vision tasks. For example, Wang et al. [10] proposed an Efficient Channel Attention (ECA), which offers an efficient relevance channel module, consisting of nonlinear adaptively determined one-dimensional convolutions to enhance attention to important information and suppress concentration on secondary information; Hou et al. [11] proposed a new Coordinate Attention (CA), which considers the influence of location information on channel information, captures location-aware information and helps the model to locate more accurately. Convolutional Block Attention Mechanism Module (CBAM [12]), which connects features in both spatial and channel dimensions in a tandem manner to generate an attention map and multiplies it with the input feature map to correct the acquired characteristics further. Based on the above study, the attention mechanism is introduced to solve the problem of inaccurate localization caused by minor target omission in the downsampling process. Therefore, this paper proposes and uses the MECA (More Efficient Channel Attention) attention mechanism, to solve the problem of high miss detection rate of small targets caused by the process of multiple downsampling of the network.

### 2.3. Feature Fusion

Feature fusion [13] is a vital tool to improve the task of target detection [14], and its purpose is mainly to merge the features extracted by the backbone network into one more discriminative element than the input image features. In various tasks such as classification, target detection, and segmentation, fusing features of different scales effectively improves performance. Low-layer feature maps have higher resolution and contain more information about location, details, etc. Still, due to fewer convolutional layers, their semantic information is lower, the perceptual field is smaller, and each pixel point of the image only extracts local details on the original image. The higher layer features have more robust semantic information, but lower resolution and poorer perception of detail information. Therefore, the effective fusion of more detailed information at the lower level and more semantic information at the higher level can improve the recall of the model, and the current commonly used feature fusion techniques include FPN [15], BiFPN [16], PANet [17], etc.

### 2.4. Training Strategies

In the model training process, various training strategies exist, and reasonable use of training strategies in real industrial deployments can lead to a better training effect of the model. In terms of model selection, larger models such as YOLOv5x and YOLOv5x6 produce better results in almost all cases. Still, their more significant number of parameters and slower inference speeds could be more favorable for mobile deployment. Therefore, this paper uses YOLOv5s [18] as the baseline training model. The anchor frame in the original YOLOv5 is calculated using K-means clustering and genetic algorithm, while the K cluster centres of the K-means clustering algorithm are chosen randomly, so the algorithm is sensitive to the initial value and is not conducive to finding the global optimal solution. Therefore, this paper adopts the K-means++ clustering algorithm [19] to reconstruct the prediction box to avoid the problem of positive and negative sample imbalance, making it more suitable for small target datasets. The CIoU [20] loss of the original network

does not consider the balance of complex and easy samples, which leads to the CIoU loss being inferior to the SIoU [21] loss in terms of detection accuracy and detection speed. Regarding the activation function, the Mish function has no upper limit, which can ensure no saturation region, so there is no problem of gradient disappearance during the training process. Therefore, in this study, we use an effective combination of Mish [22] and SIoU to replace the initial activation and loss functions.

## 3. Methods

For industrial surface defect detection, on the one hand, in order to improve the speed of model detection and make it easier to deploy on the mobile side, a new feature extraction network is proposed to enhance the original convolutional module using depth-separable convolution, which reduces the number of network parameters while maintaining little change in detection accuracy; finally, in the process of feature extraction in the backbone network, increasing the sensory field is an essential means to improve detection, while most of the studies are performed by stacking convolutional layers and introducing pooling operations to expand the perceptual domain, however, raising the perceptual field by stacking convolutional layers leads to a sharp increase in the number of parameters and a large number of pooling operations will cause more details to be lost. For this reason, we design the ASPF module to capture the multiscale feature information, by using various hole convolutions with different hole rates to reduce the loss of small target information, and introduce our proposed MECA attention module to enhance the attention to virtual channels, improve the model's extraction of multiscale detail information, and extract more detail information while obtaining a larger sensory field, and improve the model's extraction of multiscale detail information. On the other hand, since the targets to be detected are generally small, the pixels containing minor marks in the extracted feature maps become smaller and smaller as the network deepens, resulting in the model failing to achieve the expected detection effect. In contrast, the backbone network loses much detailed information while continuously downsampling, resulting in fewer small target features to be learned. Compared with the deep network, the external network is more sensitive to small targets because it retains more detailed information, so we pass the features extracted from the more shallow layer into the neck network for feature fusion to capture the exact information that is more sensitive to small targets. To address the above issues, we designed a dense multiscale feature fusion network (Dense Bidirectional Feature Pyramid Network, DBiFPN) for small target detection on industrial product surfaces to achieve the fusion of texture features and semantic features at different levels for more accurate localization of defective targets, enhanced spatial location information between adjacent feature layers. Effective cross-scale connectivity and weighted feature fusion [23] for better feature extraction.

The overall architecture of the model is shown in Figure 1, which mainly consists of three parts: backbone network, neck network, and prediction network, where the backbone network is used to extract features, and we use deep separable convolution [24] and MECA attention to design the DWC3(built from three normal convolution and multiple DWBottleneck modules) module to expand the perceptual field, remove more feature information, and reduce the number of parameters. At the end of the backbone network, we redesigned the SPPF [25] module by introducing dilate convolution to prevent the loss of more small target detail information due to pooling operation, and proposed more efficient channel attention (MECA) to enhance the awareness to virtual channels. In the neck network, we use a new multiscale weighting strategy instead of PANet to fuse more layers of features, while increasing the detail information of the image using more shallow layer features, enhancing the spatial location information between adjacent feature layers, and improving the detection accuracy of small targets. In predictive networks, the prediction layer applies the detection head [26] to the multiscale feature map of the neck network to generate detection frames and confidence levels.

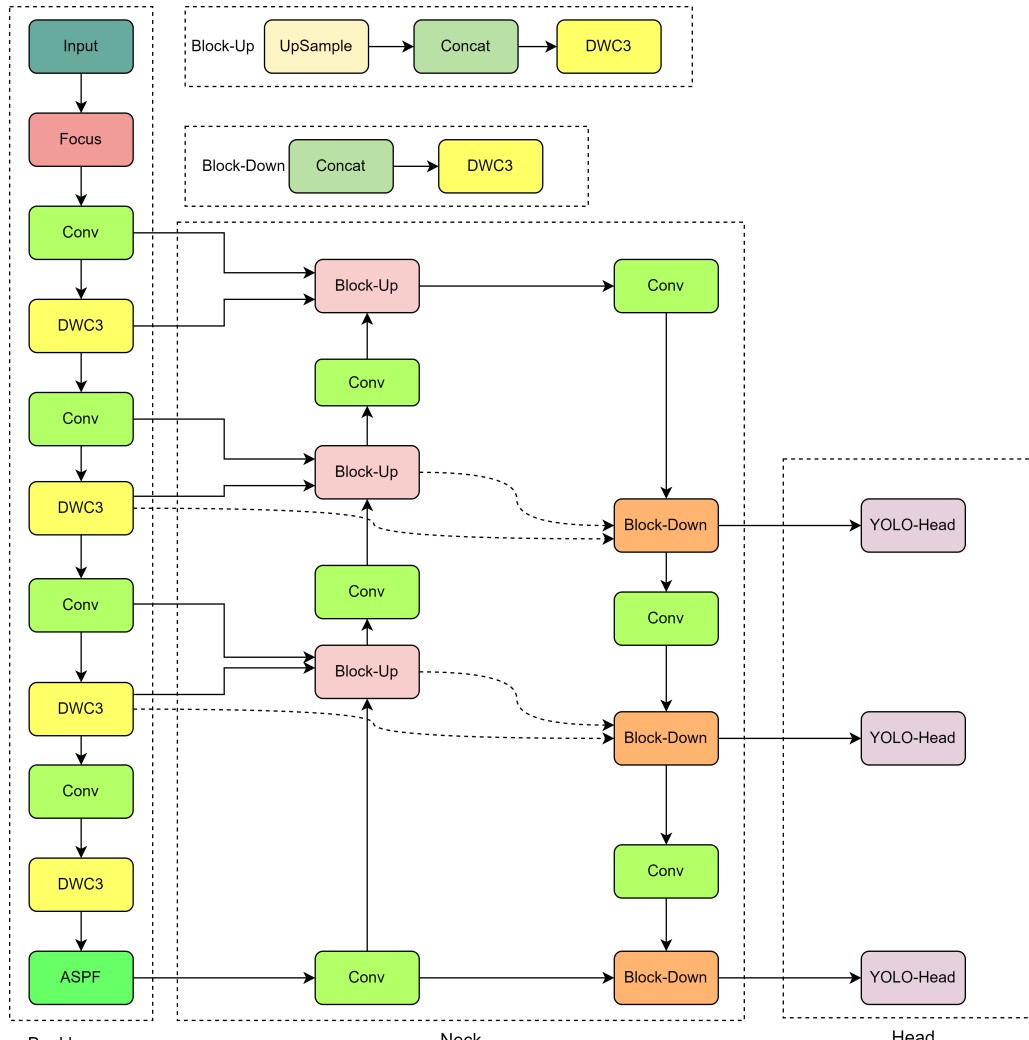

**Figure 1.** The overall architecture of the model, DWC3 forms the backbone network with a modified ASPF at the end; the neck uses dense multi-scale fusion to introduce a shallow feature map; the target is detected by three detection heads of different sizes.

### 3.1. DWC3

In this paper, to meet the needs of real industry, the conventional convolution in the C3 (contains three standard convolutional layers and several Bottleneck modules) structure is improved by using depthwise convolution and pointwise convolution. Inspired by MobileNet [27], Efficientnet [28], while maintaining the balance of detection accuracy and speed, unlike the conventional convolution, the Depthwise Convolution [29] with one convolution kernel is responsible for one channel. One channel is convolved by only one convolution kernel, followed by Pointwise Convolution [30], which puts 1x1 filter convolution on one channel to increase or decrease the depth of the feature map as a way to reduce the computational effort, but separating the channels leads to the loss of a part of the channel information, so we use the MECA channel attention module to enhance the target detection network to extract important information from the feature map and weaken irrelevant features. The improved DWC3 structure is shown in Figure 2. The CBM module consists of (Conv, Batch Normalization, Mish) and the DWBottleneck consists of (CBM, Deep Convolution, Point-by-Point Convolution, MECA Attention).

The improved DWConv and DWBottleneck are shown in Figure 3a,b, where MECA is the more efficient channel attention module.

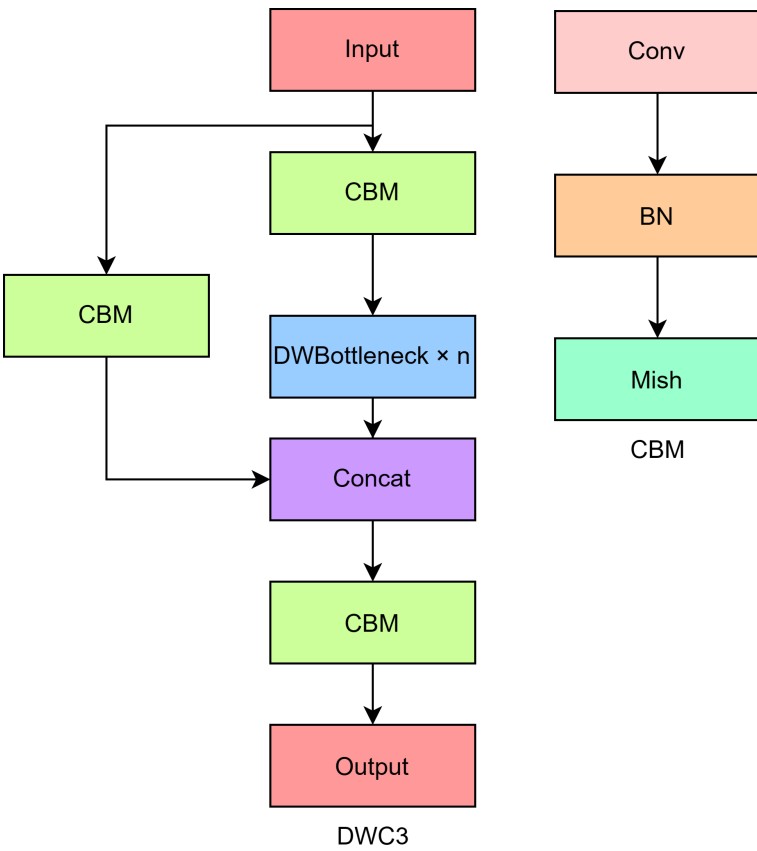

**Figure 2.** DWC3 module containing DWBottleneck.

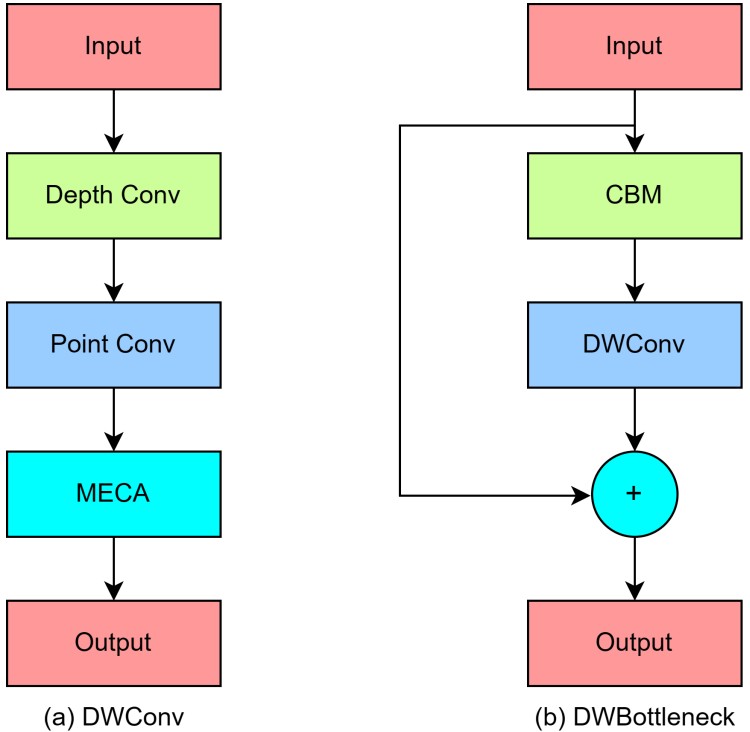

**Figure 3.** (**a**) DWConv consists of depth convolution, point-by-point convolution, and MECA attention, and (**b**) DWBottleneck consists of CBM and DWConv.

### 3.2. ASPF Module

The SPPF module in the backbone network, due to the use of multiple pooling layers, leads to the loss of a large amount of small target detail information in the extracted feature maps, especially in the case where the defective image is similar to the background, it is difficult to identify and localize the faulty target, resulting in a relatively low recall rate of the network model, inspired by the void space convolution pooling pyramid (Atrous SPP, ASPP [31]), by using the void convolution with different void rates, the obtain shallow features at different scales and retain the detailed information of the image to a larger extent, compared with the ordinary convolution operation, the dilate convolution can obtain a larger perceptual field and extract the global information of the image with the same number of parameters. Therefore, the input image features are extracted by using multiple cavity convolutions with different cavity rates when the number of parameters varies a little. To address the above problems, this paper proposes an ASPF (Atrous Spatial Pyramid Fast) module, as shown in Figure 4, to enhance contextual information by expanding neurons' receptive field and capturing high-level semantic information for target detection. First, the input image is convolved by a $1 \times 1$ convolution to reduce the number of channels and the number of censors to obtain the feature map f1, one branch of f1 passes through the channel attention MECA to obtain the weights of the channels, and the other branch will input feature f1 into the convolution kernel size of $3 \times 3$ and the dilate convolution with the dilate rate of 2, 4 and 6 to obtain feature maps f2, f3, f4 for capturing the contextual information of a larger area, then multiply feature maps f2, f3, f4 with the weights obtained from channel attention and splice them with f1 after channel attention, due to the high number of channels after splicing, and then after a $1 \times 1$ convolution to reduce the number of channels, which effectively improves the ability of the model to extract multiscale detail information and improves the model recall.

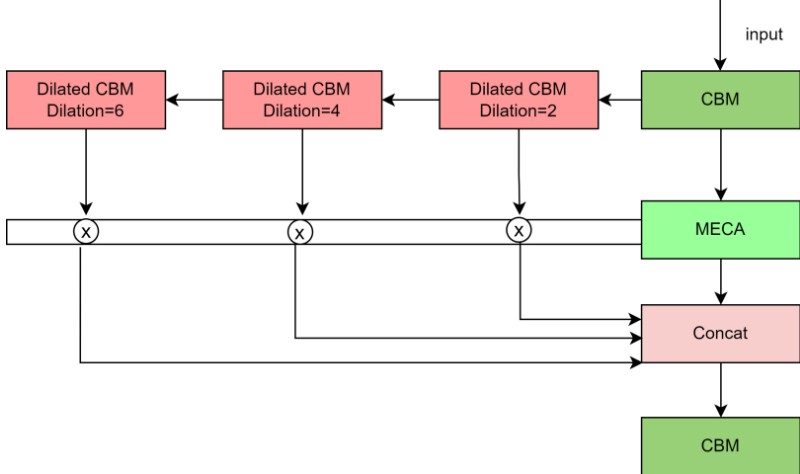

**Figure 4.** ASPF module, one branch passes through three series of dilate convolutions with dilate rates of 2, 4, and 6, one branch passes through the MECA attention module, and finally, the outputs of the two branches are stitched together.

### 3.3. DBiFPN

Since the backbone network uses a large number of convolutional and pooling layers, which leads to the loss of detail information for shallow features, making the network insensitive to small targets of industrial surface defects and leading to missed detection, and the original neck feature fusion network FPN (feature pyramid) is giving the same weight to all scales, which is not conducive to the detection of small targets, and the dataset studied in this paper is an industrial surface defect dataset, which contains many small targets, so the problems presented above are difficult to avoid. For this reason, this paper redesigns a new dense multiscale feature fusion method, which adopts a dense cross-layer cascade to fuse more layers of features, which can effectively integrate the shallow details,

edges, contours, and other information into the deeper network, which can fuse to the shallow detail information of the target with little increase in computation, so that the network can regress the target boundary more accurately and effectively improve the prediction frame and real intersection ratio of the prediction frame and the true frame. At the same time, considering that incorporating shallow features when using a cross-layer cascade will impact the deep semantic information, a weighted approach to feature fusion is used, which prevents the loss of shallow detail information to a certain extent. A dense multiscale feature fusion network is designed using the features extracted from the more shallow network (160, 160) to capture detailed information that is more sensitive to small targets (Dense Bidirectional Feature Pyramid Network ,DBiFPN), as shown in Figure 5, where P3, P4, P5, P6, and P7 are the results of the input image after downsampling by $2\times$, $4\times$, $8\times$, $16\times$, and $32\times$, respectively, and W3, W4, W5, W6, and W7 are the new feature layers after feature fusion. We use P3 with feature map size (160, 160), i.e., the more shallow layer features extracted by the backbone network are fused with the neck network, and a dense cross-layer to connect the way to obtain more detailed and spatial information. The calculation process of the Wi (i = 4, 5, 6) feature layer is detailed below; for example, W5 in Figure can be represented by the following equation.

$$W5 = \frac{W5 = [P5 \cdot w_1, P5' \cdot w_2, \text{Downsample}(P4') \cdot w_3]}{w_1 + w_2 + w_3 + \epsilon} \tag{1}$$

where Downsample denotes downsampling, Upsample denotes upsampling, [] denotes the stitching operation of different feature maps, $\epsilon$ which is a small positive number to prevent the occurrence of training instability and $w_i(i = 1, 2, 3)$ denotes the learnable weight coefficients.

$$P5' = \frac{[\text{Downsample}(P4) \cdot w_1', P5 \cdot w_{2'}, \text{Upsample}(P6') \cdot w_3']}{w_{1'} + w_{2'} + w_3' + \epsilon} \tag{2}$$

$$P4' = \frac{[\text{Downsample}(P3) \cdot w_1', P4 \cdot w_{2'}, \text{Upsample}(P5') \cdot w_3']}{w_{1'} + w_{2'} + w_3' + \epsilon} \tag{3}$$

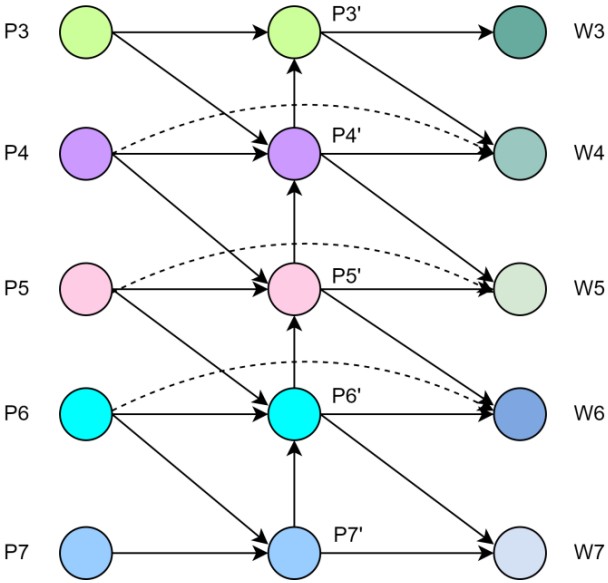

**Figure 5.** DBiFPN, fusing more detailed information in the feature map using the more shallow feature map P3 (160, 160), fusing more feature information using cross-layer connectivity.

In the feature fusion process, the top and bottom node information undergoes less convolution and retains more detailed information. To reduce the complexity of the model,

the cross-layer connection is no longer used for feature fusion in the top and bottom nodes, for example, W7 can be obtained by the following equation.

$$W7 = \frac{\left[\, \text{Downsample} \, (P') \cdot w_1, P' \cdot w_2\right]}{w_1 + w_2 + \epsilon} \tag{4}$$

$$P6' = \frac{\left[\, \text{Downsample} \, (\, P5 \,) \cdot w_1', P6 \cdot w_2, \, \text{Upsample} \, (P7') \cdot w_3'\right]}{w_1' + w_{2'} + w_3' + \epsilon} \tag{5}$$

$$P7'' = \frac{\left[\, \text{Downsample} \, (P6) \cdot w_1', P7 \cdot w_{2'}\right]}{w_{1'} + w_{2'} + \epsilon} \tag{6}$$

### 3.4. MECA Attention

The MECA (More Efficient Channel Attention) module proposed in this paper considers that the maximum pooling process on the input feature map can extract detailed information, such as texture features on the feature map, and the average pooling process can extract contextual features related to the target on the feature map. Therefore, the average pooling and maximum pooling operations are performed simultaneously on the input feature map, and the stitching operation is performed on the feature maps of the two channels after processing. The overall flowchart of the MECA module is shown in Figure 6 below, and the computational process can be expressed as follows.

$$F' = \text{cat}[\text{AvgPool}(F), \text{MaxPool}(F)] \tag{7}$$

$$\omega_c = \sigma\big(\text{ASK}\big(\text{Conv}\big(F'\big)\big)\big) \tag{8}$$

$$F'' = \omega_c \otimes F \tag{9}$$

where $F$ denotes the input feature map, AvgPool and MaxPool denote the average pooling and maximum pooling, respectively, $w_c$ denotes the learnable weights, cat denotes the splicing operation, ASK denotes the one-dimensional convolution of adaptively selected k, $\otimes$ denotes the feature map by bit multiplication, and F'' denotes the output feature map.

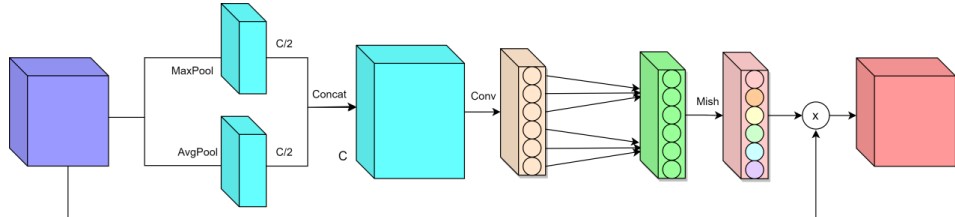

**Figure 6.** MECA Attention Module.

### 3.5. Loss Function

When solving target detection problems with convolutional neural networks, whether they do classification or regression problems, the loss function is indispensable [32] . The loss function of the first-order detector in target detection consists of three parts, which are confidence loss Lobj, classification loss Lcls and regression loss Lloc of the prediction frame. The original YOLOv5 classification loss and target loss used BCE Loss (cross entropy loss), and the localization loss used CIoU Loss. while the CIoU loss function aspect ratio describes the relative value, there is a certain ambiguity. In addition, as the imbalance between positive and negative samples of the first stage target detector is particularly prominent, the number of negative samples is much higher than that of positive samples, thus making negative samples dominate the loss and ultimately leading to poorer model training, while CIoU does not consider the balance of difficult and easy samples, which leads to CIoU loss being inferior to SIoU loss in terms of detection accuracy and detection speed. SIoU considers the vector angle between the real frame and the predicted frame

and redefines four losses, including angle loss, distance loss, shape loss, and IoU loss. Its calculation formula is shown as follows.

$$\text{LOSS}_{\text{SIoU}} = 1 - \text{IoU} + \frac{\Delta + \Omega}{2} \tag{10}$$

where IoU denotes the degree of overlap between the predicted frame and the rear frame, $\Delta$ denotes the distance loss between the predicted frame and the rear frame, $\Omega$ denotes the shape loss between the predicted frame and the real frame, and the IoU loss can be expressed by the following equation:

$$\text{IoU} = \frac{A \cap B}{A \cup B} \tag{11}$$

where A denotes the area of the predicted bounding box, B denotes the area of the real ground box, $\cap$ used to find the overlapping area between the two anchor boxes A and B, $\cup$ used to find the common area between the two anchor boxes A and B. The distance loss between the predicted box and the real box can be expressed by the following formula, from which it can be seen that when $\wedge$ tends to 0, $\Delta$ the contribution is greatly reduced, and as the angle $\wedge$ increases, $\Delta$ the contribution becomes larger, and the problem solving becomes more and more difficult. Therefore, as the angle increases, the distance value is obtained by time priority.

$$\Delta = \sum_{t=x,y} \left(1 - e^{-\gamma p t}\right) = 2 - e^{-\gamma \rho_x - e^{-\gamma \rho_y}} \tag{12}$$

$$\rho_x = \left(\frac{b_{c_x}^{gt} - b_{c_x}}{c_w}\right)^2, \rho_y = \left(\frac{b_{c_y}^{gt} - b_{c_y}}{c_h}\right)^2, \quad Y = 2 - \Lambda \tag{13}$$

where $c_w$, $c_h$ are the width and height of the smallest external rectangle of the real frame and the predicted frame, respectively, $b_{c_x}^{gt}$, $b_{c_y}^{gt}$ are the center coordinates of the real frame, $b_{c_x}$, $b_{c_y}$ are the center coordinates of the predicted frame, and $\wedge$ denotes the angular loss between the predicted frame and the real frame, which can be expressed by the following equation.

$$\Lambda = \cos\left(2^* \left(\arcsin\left(\frac{c_h}{\sigma}\right) - \frac{\pi}{4}\right)\right) \tag{14}$$

$$\sigma = \sqrt{\left(b_{c_x}^{gt} - b_{c_x}\right)^2 + \left(b_{c_y}^{gt} - b_{c_y}\right)^2} \tag{15}$$

$$c_h = \max\left(b_{c_y}^{gt}, b_{c_y}\right) - \min\left(b_{c_y}^{gt}, b_{c_y}\right) \tag{16}$$

where $\sigma$ is the distance between the center point of the real frame and the predicted frame, $c_h$ and is the height difference between the center point of the real frame and the predicted frame, and the shape loss can be expressed by the following equation.

$$\Omega = \sum_{t=w,h} \left(1 - e^{-w_t}\right)^\theta = \left(1 - e^{-w_w}\right)^\theta + \left(1 - e^{-w_h}\right)^\theta \tag{17}$$

$$W_w = \frac{\left|w - w^{gt}\right|}{\max\left(w, w_{gt}\right)}, W_h = \frac{\left|h - h^{gt}\right|}{\max\left(h, h^{gt}\right)} \tag{18}$$

where w and h are the width and height of the predicted frame, respectively, and $w^{gt}$, $h^{gt}$ is the width and height of the real frame, $\theta$ used to control the degree of attention to shape loss.

## 4. Experiment

In this paper, two industrial surface defect datasets are utilized to evaluate the performance of our proposed model. Firstly, the NEU-DET steel dataset is used to verify the effectiveness of the proposed innovation points through a large number of ablation experiments and comparative experiments, and secondly, the generalization of our model is verified by the PCB dataset.

### 4.1. Experimental Setup

#### 4.1.1. Experimental Parameters

This paper's experimental study and network training are run in a lab server environment with torch1.9.1, CUDA11.0, CUDNN11.3, and NVIDIA-A100 graphics cards. To ensure the accuracy and reliability of the ablation experiment, the same hyperparameters need to be set. In this paper, The input image size is $640 \times 640$, Batch-size is 32, Num-workers is 8, the number of cycles is 300 rounds, the learning rate is 0.01, and the momentum is 0.937.

#### 4.1.2. Datasets

The first dataset is the NEU-DET steel surface defect dataset published by Northeastern University, and the dataset connection obtained is http://202.118.1.237/yunhyan/NEU-surface-defect-database.html (accessed on 1 September 2022). The baseline of this dataset contains 1800 pictures, and 3600 pictures are obtained after data processing, including six defect categories: cracks, inclusions, patches, surface pockmarks, rolled scales and scratches. Each defect category contains 600 images, each image is $200 \times 200$ pixels in size, and the training and validation sets are divided into 9:1. Figure 7 shows examples of different defects in the baseline. From the grayscale map, it can be seen that the same defect may have large differences in appearance, for example, scratch images may contain horizontal scratches and vertical scratches.

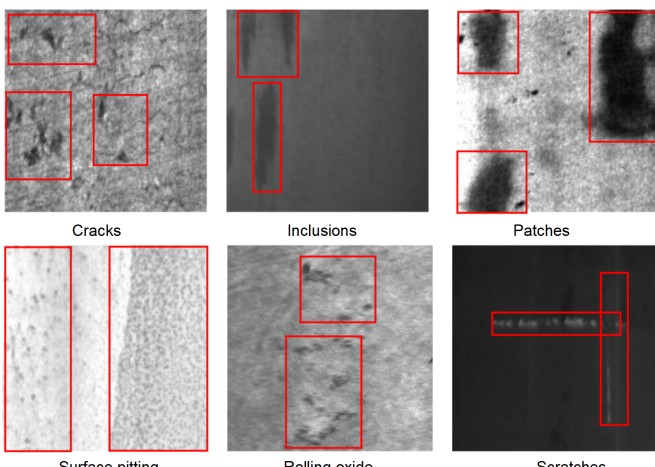

**Figure 7.** Visualization of steel dataset images, with red boxes representing defective positions.

The second dataset is PCB, and the PCB board is a PCB dataset published by the Intelligent Robotics Open Laboratory of Peking University, and the connection of this dataset is GitHub-YMkai/PCB_Datasets: Two PCB datasets. The baseline of the dataset contains 693 pictures, and 1780 pictures are obtained after data processing, with a total of 6 defect types, namely missing holes, rat bites, open circuits, short circuits, burrs, and residual copper, and then the training set and the verification set are divided into 9:1. Figure 8 below shows an example of a defect in the dataset baseline. It can be seen from the picture that the proportion of defective images is very small compared to the whole image, and an image contains multiple defects, which increases the difficulty of defect detection to a certain extent.

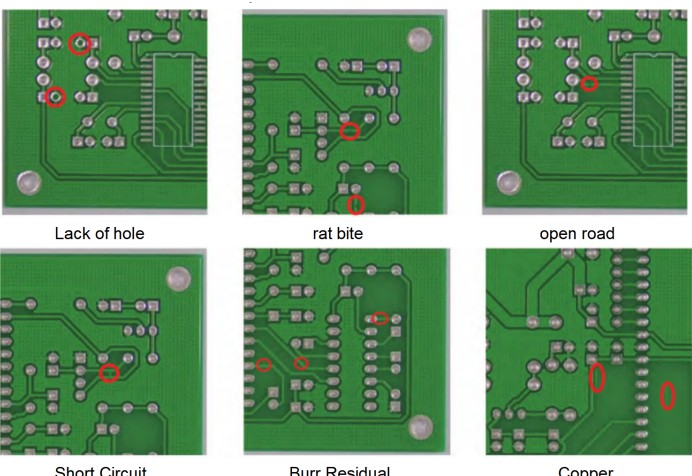

**Figure 8.** Visualization of images of six types of defects in PCB boards, and the red box represents the defective position.

4.1.3. Evaluation Metrics

To validate the performance of the proposed model, Precision (precision), Recall (recall), mAP (mean accuracy), F1 (area enclosed by PR curves), and FPS (model inference speed) are used as measures for model evaluation, where Recall, mAP are used as the main evaluation metrics, and specifically, mAP is the average of all classes of all 10 intersection points in the range [0.50, 0.95] averaged over the joint (IoU) threshold. the definition of Recall, mAP is shown below.

$$R = \frac{T_P}{T_P + F_N} \tag{19}$$

$$mAP = \frac{\sum_{i=1}^{C} AP_i}{C} \tag{20}$$

where $T_P$ refers to true positives, i.e., the predicted value is true and the actual value is also true, $F_N$ and false positives, i.e., the predicted value is true and the actual value is false.

Using precision and recall alone cannot objectively reflect the goodness of the model, therefore, F1, which combines both evaluation metrics, is used in the paper to measure the performance of the model. F1 refers to the area enclosed by the P-R curve, the larger the area, the better the performance of the model; in addition, in practical industrial applications, detection speed is also an important index, and we use FPS to evaluate the inference speed of the model. F1 can be expressed by the following equation.

$$F1 = 2 \times \frac{P \times R}{P + R} \tag{21}$$

$$P = \frac{T_P}{T_P + F_P} \tag{22}$$

where $F_P$ refers to false negatives, where the predicted value is negative and the actual value is true.

*4.2. Data Pre-Processing*
4.2.1. Data Augmentation

The two datasets used in this paper are small because of the amount of data, the model may not converge in the training process and fail to achieve the expected results, and the images taken in the dataset are all regular rectangles. However, in the actual use of these industrial products may be disturbed by tilt, lighting or other environmental factors, so the paper uses random data augmentation, the original dataset of images are randomly rotated,

panned, cropped, the brightness of the image changes and other random augmentation methods to obtain the expanded dataset as shown in Figures 9 and 10.

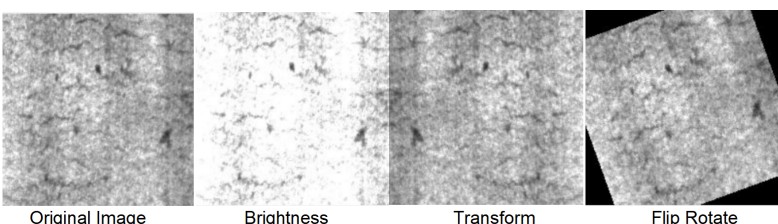

| Original Image | Brightness | Transform | Flip Rotate |

**Figure 9.** Enhanced images of the steel dataset.

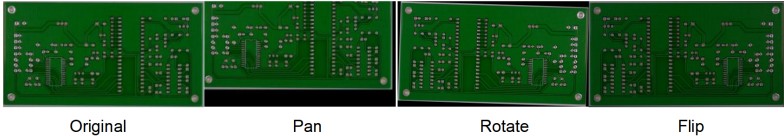

| Original | Pan | Rotate | Flip |

**Figure 10.** Enhanced image of PCB dataset.

### 4.2.2. K-means++

An appropriate anchor frame can reduce the loss value and computational effort and improve the speed and accuracy of target detection. Since the anchor frames in the original YOLOv5 were calculated using K-means clustering and genetic algorithms, and the K cluster centres of the K-means clustering algorithm were randomly selected so the algorithm is sensitive to the initial value. It could be more conducive to finding the optimal global solution. To address the above problems, we propose and adopt an improved K-means++ clustering algorithm to reset the anchor frame. K-means++ algorithm steps are shown in Algorithm 1.

---

**Algorithm 1** Steps of the K-means++ Clustering Algorithm.

---

Input: labels of the training dataset.
Output: the values of k anchor boxes
(1) Randomly select a point from the data set as the first clustering center.
(2) Calculate the shortest distance D(x) = 1-IOU(box, center) between each sample and the current nearest cluster center.
(3) Calculate the probability that each sample is selected as the next clustering center
$P(x) = \frac{D(x)^2}{\sum_{j=1}^{n} D(x_j)^2}$.
(4) Repeat steps (2) and (3) until k clustering centers are found.
(5) Calculate the distance from each sample in the training set to the k cluster centers and assign it to the cluster corresponding to the cluster center with the closest distance.
(6) Reset the centers of each class.
(7) Repeat steps (5) and (6) until the cluster center positions no longer change.

---

### 4.3. Steel Dataset Ablation Experiment

#### 4.3.1. Comparison of Backbone Networks

In this part of the exposition, the recall (R), mean accuracy (mAP), and frames per second (FPS) of the model will be used for the comprehensive evaluation of the defect detection model. To verify the effectiveness of the proposed one-stage target detection model in this paper, the performance of the proposed new feature extraction network is first compared by comparing different network structures with the original YOLOv5s model, and the original CSPDarknet53 backbone network is compared with the proposed new feature extraction network, as shown in Table 1.

As can be observed from the experimental data in Table 1, the improved DWC3 structure using depth-separable convolution and more efficient channel attention enables the network to focus more on the features of important channels while reducing the number

of model parameters. It improves R by 1.4%, mAP by 1.3%, and F1 by 1.4% compared to the original network model. The ASPF structure proposed in this paper retains more image detail information, which can improve the model's contextual information extraction ability while enhancing the feature map perceptual field, enhances the focus on important channels, improves the model's extraction of multiscale detail information, and avoids the loss of small target detail information caused by pooling operations. Compared with the original YOLOv5s network model, R improves by 3.7%, mAP improves by 4.4%, F1 improves by 2.76%, and the number of parameters increases slightly. In the case of the DWC3 structure combined with the ASPF structure, although the detection speed is lower than the original algorithm, it can meet the needs of practical industrial detection. At the same time, R improves by 5.1%, mAP by 4.7%, and F1 by 3.36%.

**Table 1.** Effect of different network structures on the detection effect of the model.

| Model | R(%) | mAP@0.5 (%) | F1 (%) | Params (M) | FPS |
|---|---|---|---|---|---|
| YOLOv5s + DWC3 | 84.2 | 88.9 | 85.23 | 67.15 | 61 |
| YOLOv5s + ASPF | 86.5 | 92 | 86.59 | 73.52 | 51 |
| YOLOv5s + DWC3 + ASPF | 87.9 | 92.3 | 87.19 | 72.13 | 54 |

#### 4.3.2. Comparison of Different Neck Networks

Comparing the effects of the DBiFPN feature extraction network with PANet and BiFPN networks on model detection accuracy, Table 2 shows the effects of different feature extraction networks on the model detection effect.

**Table 2.** Shows the effects of different Neck networks on model detection accuracy.

| Neck Network | R (%) | mAP@0.5 (%) | F1 (%) | Params (M) |
|---|---|---|---|---|
| FPN + PANet | 82.8 | 87.6 | 83.83 | 70.26 |
| BiFPN | 83.1 | 87.9 | 83.95 | 80.93 |
| DBiFPN | 83.9 | 88.9 | 84.46 | 80.92 |

From the experimental data in Table 2, we can observe that DBiFPN outperforms the PANet and BiFPN network in terms of detection accuracy. Compared with PANet and BiFPN, DBiFPN improves 1.1% and 0.8% in R, 1.3% and 1.0% in mAP, and 0.63% and 0.51% in F1, respectively, in this paper designed a multiscale feature fusion network to capture detailed information that is more sensitive to small targets using features fused to the neck network at more shallow layers and to connect feature maps from different layers with cross-layer weighting to predict classification information and bounding boxes to reduce information loss in defective images.

#### 4.3.3. Comparison of Attention Modules

This subsection compares the effects of the ECA attention module and the MECA attention module proposed in this paper on the network detection effect. It can be seen from the experimental results in Table 3 below that the MECA attention module improves R by 2%, mAP by 0.7%, and F1 by 0.44% compared to the ECA module with little change in the number of parameters. By analyzing the experimental results, it is concluded that maximum pooling in MECA can extract detailed information, such as texture features, on the feature map. Average pooling processing can extract contextual features related to the target on the feature map, so processing the input feature map in both ways can make the network pay more attention to important information in the feature map and locate the defective target more accurately.

### 4.3.4. Comparison of Anchor Frame Selection

In YOLOv5, the anchor box selection is obtained by the K-means clustering algorithm. Since the K clustering centers of the K-means clustering algorithm are randomly selected, the K-means algorithm is sensitive to the initial value. It is random, which could be more conducive to finding the optimal global solution, so we use the improved K-means++ clustering algorithm to reset the anchor frame size, and the relationship between the number of anchor frames k and IOU can be seen in Figure 11; when the number of anchor frames is approximately equal to 9, the IOU starts to converge regionally, so we choose k = 9 and obtain the new anchor frames as shown in Table 4 below.

**Table 3.** Effect of different attention modules on model detection results.

| Model | R (%) | mAP@0.5 (%) | F1 (%) | Params (M) |
|---|---|---|---|---|
| YOLOv5s | 82.8 | 87.6 | 83.83 | 70.26 |
| YOLOv5s + ECA | 84.4 | 88.9 | 84.79 | 70.29 |
| YOLOv5s + MECA | 86.4 | 89.6 | 85.23 | 70.30 |

**Table 4.** The relationship between the selection of anchor frame and feature graph.

| Prediction | Feature Map Size | Original Anchor Frame | K-means++ Anchor Frame |
|---|---|---|---|
| Big target | 20 × 20 | 10 × 13, 16 × 30, 33 × 23 | 11 × 12, 17 × 18, 15 × 23 |
| Medium target | 40 × 40 | 30 × 61, 62 × 45, 59 × 119 | 24 × 28, 27 × 53, 42 × 21 |
| Small goal | 80 × 80 | 116 × 90, 156 × 198, 373 × 326 | 40 × 35, 65 × 74, 127 × 134 |

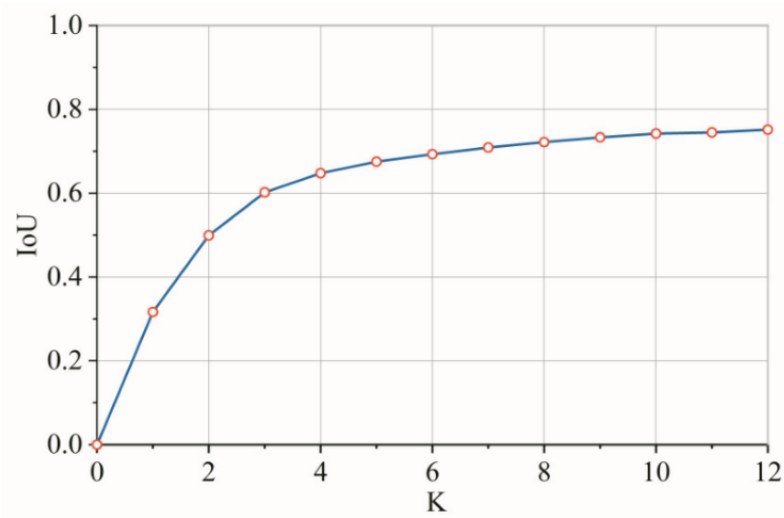

**Figure 11.** Variation curve of IOU with anchor frame k.

The comparative results from the experiments in Table 5 show that using the K-means++ clustering algorithm improves R by 0.7%, mAP by 1.1%, and F1 by 0.29% compared to the original network model. It shows that the K-means++ algorithm as an optimized clustering center enhances the localization and improves the detection accuracy of the algorithm.

**Table 5.** Effect of K-means++ clustering algorithm on model detection results.

| Model | R (%) | mAP@0.5 (%) | F1 (%) | Params (M) |
|---|---|---|---|---|
| YOLOv5s | 82.8 | 87.6 | 83.83 | 70.26 |
| YOLOv5s + K-means++ | 83.5 | 88.7 | 84.12 | 70.38 |

### 4.3.5. Comparison of Different Loss Functions and Activation Functions

For different activation functions and loss functions, the detection performance varies for different defect classes. Therefore, this paper uses the current popular loss and activation functions to compare the model detection performance. We have selected SiLU, Mish, and ReLU activation functions with CIoU and SIoU considered functions for comparison experiments. The experimental results are shown in Table 6 below.

**Table 6.** Effects of different activation functions and loss functions on the same model.

| Model | Activation/Loss Function | R (%) | mAP@0.5 (%) | F1 (%) | FPS |
|---|---|---|---|---|---|
| YOLOv5s | SiLU/CIoU | 82.8 | 87.1 | 82.89 | 52 |
| YOLOv5s | ReLU/CIoU | 80.5 | 87.3 | 82.54 | 52 |
| YOLOv5s | Mish/CIoU | 83 | 87.6 | 82.39 | 52 |
| YOLOv5s | SiLU/SIoU | 83.8 | 87.7 | 82.91 | 52 |
| YOLOv5s | ReLU/SIoU | 82.8 | 87.6 | 82.90 | 52 |
| YOLOv5s | Mish/SIoU | 83.6 | 88.2 | 83.95 | 52 |

From the experimental results in Table 6, it can be seen that the combination of the Mish activation function and SIoU loss function obtained the highest mAP, compared with the combinations of the other five, mAP improved by 1.1%, 0.9%, 0.6%, 0.5%, 0.6%, F1 improved by 1.06%, 1.41%, 1.56%, 1.04%, 1.05%, R was only second to The combination of SiLU and SIoU, from the table, we can conclude that SIoU is better than CIoU in detection accuracy, SiLU and ReLU are faster than Mish, but the accuracy is worse, so the combination of Mish and SIoU is finally chosen.

### 4.4. Steel Dataset Comparison Experiment

In this paper, we analyze the performance of the model proposed in this paper from several perspectives, where mAP (mean average precision) is used as the main measure of model accuracy, and FPS is used as a measure of model inference speed. By analyzing the experimental results in Figure 12, we can observe that our model is higher in mAP metrics compared to the one-stage detection models YOLOv3s, YOLOv4s, YOLOv5s, YOLOX, YOLOv7s, and the two-stage detection model Faster R-CNN, and is equal to the fastest YOLOv7s in terms of FPS inference speed.

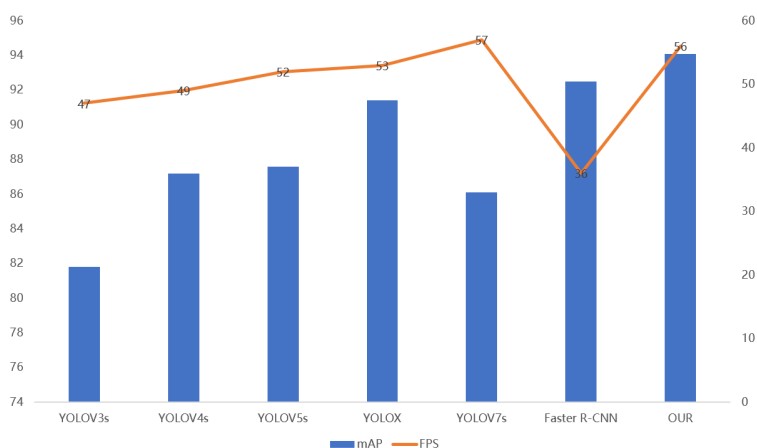

**Figure 12.** In the comparison of mAP and FPS values of different detectors, the horizontal axis represents different detectors, the vertical axis in the left half represents mAP values, and the vertical axis in the right half represents FPS values.

In this section, we tested the model's detection accuracy, speed, and model complexity and compared the experiments with the other six SOTA detectors YOLOv3s, YOLOv4s, YOLOv5s, YOLOX, YOLOv7s, and Faster R-CNN. The results are shown in Table 7 below.

The experiments verified the effectiveness of our model. Compared with the other five detectors, mAP improved by 12.3%, 6.9%, 6.5%, 2.7%, 8.0%, and 1.6%, respectively, and F1 improved by 8.44%, 5.32%, 5.74%, 0.97%, 6.44%, and 1.45%, respectively. However, it is slightly lower than the other models in terms of detection speed; it can meet the actual industrial demand.

**Table 7.** Comparison of different detector models on the steel dataset.

| Model | R (%) | mAP@0.5 (%) | F1 (%) | Params (M) | FPS |
|---|---|---|---|---|---|
| YOLOv3s | 82.3 | 81.8 | 81.13 | 93.14 | 47 |
| YOLOv4s | 83.6 | 87.2 | 84.25 | 91.2 | 49 |
| YOLOv5s | 82.8 | 87.6 | 83.83 | 70.26 | 52 |
| YOLOX | 86.6 | 91.4 | 88.6 | 75.45 | 53 |
| YOLOv7s | 84.2 | 86.1 | 83.13 | 91.48 | 57 |
| Faster R-CNN | 88.4 | 92.5 | 88.12 | 552.5 | 36 |
| OUR | 90.2 | 94.1 | 89.57 | 90.12 | 56 |

*4.5. Visualization Research*

Figure 13 below shows the visualization of the detection results of different detectors on the steel dataset. Since the steel surface has oxide layers of different colors and shapes, it is difficult to identify the background and defect images. The target defects are irregular with different shapes of the same type of defects; we chose four types of defects in the steel dataset that are more difficult to detect: cracks, inclusions, paper-tie oxide, and scratches categories for model visualization study. From the figure below, it can be seen that YOLOv7s, YOLOv3s, YOLOv4s, and YOLOv5s all have leakage in detecting cracks with a large number of small targets, and YOLOX has leakage in identifying scratches with a large change in shape. The confidence score is higher.

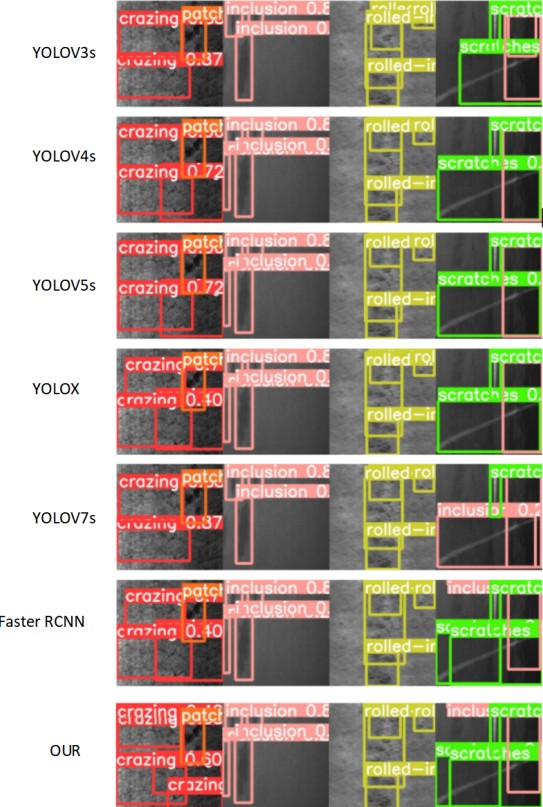

**Figure 13.** Visualization of different detectors for detecting images on steel dataset.

### 4.6. Comparison Experiments of Different Detectors for PCB Datasets

In this section, the PCB dataset is used to train on the proposed model in this paper, and to compare it with other detectors to demonstrate the generalization ability of our proposed model. By analyzing the mAP values in Figure 14, it can be observed that our model is superior to YOLOv3s, YOLOv5s, YOLOv7s, and Faster R-CNN. Although the FPS of our model is slightly lower than that of YOLOv7s, the mAP values are much improved.

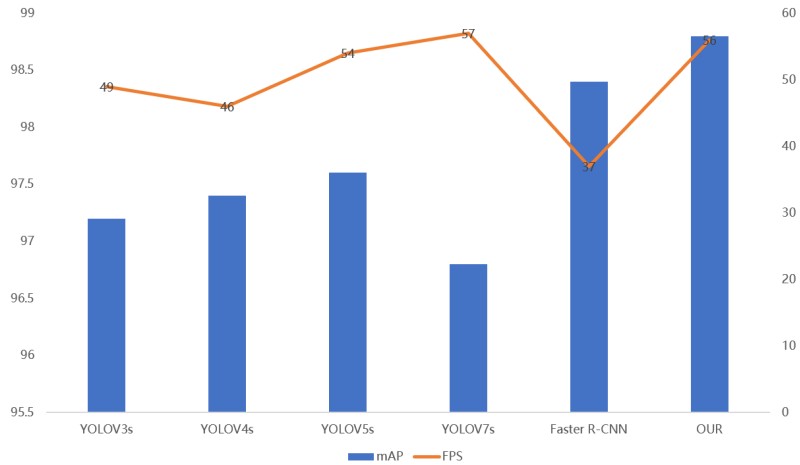

**Figure 14.** In the comparison of mAP and FPS values of different detectors, the horizontal axis represents different detectors, the left half of the vertical axis represents mAP values, and the right half of the vertical axis represents FPS values.

As can be seen from the experimental results in Table 8 below, the generalization ability of the proposed model in this paper is demonstrated on the PCB dataset. mAP is improved by 1.6%, 1.4%, 1.2%, 2.0%,F1 is improved by 0.95%, 0.86%, 0.93%, compared to YOLOv3s, YOLOv4s, YOLOv5s, YOLOv7s and other models, respectively. The 1.44% test, although it is lower than Faster R-CNN in F1 value, the number of Faster R-CNN parameters as a two-stage target detector is much higher than that of the one-stage detector, and the inference speed is slower, which does not apply to the actual industrial needs.

**Table 8.** Comparison of different detector models on the PCB dataset.

| Model | R (%) | mAP@0.5 (%) | F1 (%) | Params (M) | FPS |
|---|---|---|---|---|---|
| YOLOv3s | 96.5 | 97.2 | 96.59 | 85.4 | 49 |
| YOLOv4s | 95.5 | 97.4 | 96.68 | 91.25 | 46 |
| YOLOv5s | 95.7 | 97.6 | 96.61 | 70.26 | 54 |
| YOLOv7s | 95.2 | 96.8 | 96.1 | 91.48 | 57 |
| Faster R-CNN | 96.5 | 98.4 | 97.58 | 552.5 | 37 |
| OUR | 96.9 | 98.8 | 97.54 | 90.12 | 56 |

### 5. Discussion

Table 1 shows the results of different improved structures in the backbone network for ablation experiments. The combination of lightweight DWC3 and ASPF modules improves the mAP value by 4.7% compared to the original YOLOv5 with little increase in the number of parameters; Table 2 shows the effect of different neck networks on model detection. The DBiFPN proposed in this paper uses more shallow features fused to the neck network to capture Table 3 shows the effect of different attention modules on model detection, and the MECA attention designed in this paper improves 0.7% in mAP over the ECA attention; Table 5 discusses the effect of reconstructing the anchor frame using the K-means++ clustering algorithm on the detection results, and compared to the original anchor frame size, this algorithm improves 1.1% in mAP value is improved by 1.1%; Table 6

shows the effect of different activation and loss functions on the model, and the optimal combination strategy is chosen to achieve the best inference for the model; by comparing Table 7 with different detectors, it can be observed that our method achieves satisfactory results on the steel dataset, and the detection speed is slightly lower than the current faster YOLOv7; to verify the effectiveness of our proposed model and generalizability, the PCB dataset is used in Table 8 for comparison with other detectors.

## 6. Conclusions and Future Outlook

This paper proposes an industrial defect detection method based on an expanded sensory field and feature fusion. To verify the effectiveness of the proposed model, we use two industrial datasets for testing. Firstly, a new feature extraction network is designed using deep separable convolution and MECA attention to reduce the computational effort. A dilate convolution with different dilate rates is introduced at the end of the backbone network to expand the perceptual field and extract more semantic information. Secondly, a new dense multiscale weighting strategy is proposed in the neck network to introduce more shallow layer feature maps to increase the detailed information of images, enhance the spatial location information between adjacent feature layers, and improve the detection accuracy of small targets. Finally, this paper uses some training strategies for model optimization, reconstructs the prediction frame using the K-means++ algorithm to find the best anchor frame for small target datasets, and verifies the effectiveness of the combination of the Mish activation and the SIoU loss function. Compared with other industrial surface defect detection algorithms, this algorithm outperforms other steel surface detection algorithms.

Although the method proposed in this paper improves detection accuracy and speed, there are still some issues to be discussed. The dense multiscale feature fusion network proposed in this paper achieves some detection results, but exploring a more lightweight and efficient feature fusion network is necessary.

**Author Contributions:** Project administration, G.L.; data curation, G.L.; writing—original draft, S.Z.; writing—review and editing, R.S. and Z.Z.; funding acquisition, M.L. and M.Z.; resources, D.H. All authors have read and agreed to the published version of the manuscript.

**Funding:** This work was supported by the Taishan Scholars Program(NO.tsqn202103097),and the Key R & D plan of Shandong Province (Soft Science Project)(2022RZB02012).

**Data Availability Statement:** The data used to support the findings of this study are available from the corresponding author upon request.

**Acknowledgments:** The authors would like to thank all the anonymous reviewers for their insightful comments and constructive suggestions.

**Conflicts of Interest:** The authors declare no conflict of interest.

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
