# Peer review of "YOLO-RFF: An Industrial Defect Detection Method Based on Expanded Field of Feeling and Feature Fusion"

_electronics, doi:10.3390/electronics11244211_

Round 1

Reviewer 1 Report

In this work, Authors have presented a new feature extraction network is proposed to improve the original convolution module by using depth-separable convolution to reduce the computational effort based on its feature extraction capability. Also for catering the need of resolving the problems of low efficiency, high false detection rate and poor real-time performance of current industrial defect detection methods, this paper proposes an industrial defect detection method based on expanded perceptual field and feature fusion for practical industrial applications. First, in order to improve the real-time performance of the network, the original network structure is improved by using depth-separable convolution to reduce the computation while ensuring the detection accuracy, and the important information extraction from the feature map is enhanced by using MECA attention to the detection network. In order to reduce the large amount of loss of small target detail information caused by pooling operation, the ASPF module is constructed by using null convolution with different null rates to extract more contextual information. Secondly, to improve the detection accuracy, a new feature fusion method is proposed to fuse more detail information by introducing a shallower feature map and using a dense multi-scale weighting method. Finally, in the process of model optimization, the k-means++ algorithm is used to reconstruct the prediction frame to speed up the convergence of the model as well as to verify the effectiveness of the combination of the Mish activation function and the SIoU loss function.

The reviews are as follows:

The authors present very nice work with detailed descriptions and sufficient literature.

There are a few typographical errors, like in line number 139.

The caption in Figure must be written properly.

What is the significance of the expression mentioned in equation 12. Authors must justify the importance of the parameters used.

Authors must provide citation details for the dataset taken from the database.

Author Response

First of all, thank you for giving some guiding comments in your busy schedule, the manuscript has been revised one by one according to your comments, please refer to the attachment.

Reviewer 2 Report

The manuscript presents an improved YOLO object detection model aimed at real-time defect detection in the electronics industry, and trained with two datasets: surfaces defect in the steel industry and defective PCB images.

The authors propose a number of changes (from recent literature) in the architecture of a baseline model that improves its performance against several object detection models like yolov3, yolov4, yolov5, yolov7 or faster RCNN, which is a representative set of current OD algorithms. The proposed changes try to improve detection metrics, speed, detection of small defects and a better foreground detection, which is relevant for the proposed field of application. The performance gain on two industry-related datasets is clear from the extensive experimental details, and in this regards, I believe this work is interesting for the readers of Electronics journal.

My main concern is about readability of the manuscript, in particular, in the introduction.  Also, there are multi-line sentences hard to read, like the  sentence in 3.1 (lines 196-206), 11 lines long.  Or the first paragraph in section 3.5. I strongly suggest the manuscript to be proof-read for readability.

I have the following specific comments:

1.       I believe that an “and” is missing in “Steel dataset PCB dataset” in the abstract.

2.       The start of the introduction should be rewritten, it needs some context. For example, the expression “.. and other aspects of the impact of the rolling process…” in just the third line of the manuscript is confusing: which specific industry are your referring to?  Why referring to the rolling process? It is relevant, for example, to PCB manufacturing?  If the argument is relevant to the generic field of manufacturing, it should be more general; if is relevant to a specific process and/or industry, it should be stated.

3.       I think this phrase in the first paragraph of introduction needs to be rewritten: “Currently, industrial research based on deep learning can update parameters in real-time based on training data with high…” I don’t understand why real-time could be relevant in the training phase, or even if the term “real-time” during training has any meaning. Or are they referring by “parameters” to an industrial process and not to the model’s parameters?

4.       The phrase “, Guo et al [4] proposed a Transformer improved YOLOv5-based steel surface defect detection method, which achieved some results in terms of detection accuracy, but its achieved certain results, but its detection speed could not meet the actual industrial demand” is unreadable.

5.       2.1. Feeling the Wild”

6.       There are two “Finally” in lines 154 and 155.

7.       The baseline model for the entire work is yolov5. It must be properly credited (I believe it is https://github.com/ultralytics/yolov5) , reference [18] does not seem adequate to me.

8.       MECA meaning should be stated in the first appearance (it is not defined until line 273). There are many acronyms in the manuscript that are not explained, however, it could be obvious to machine learning practitioners, but not probably  to all readers of this journal (such as DWC, DBiFPN, SPPF…)

9.       There is section (3.2) devoted to ASPF but it is not defined.

10.   I believe that the proposed MECA should be put in context with previous works in 2.2.

11.   There are references to void convolutions and  null convolutions, and to dilated convolutions and hole convolutions, my suggestion is to unify naming.

12.   CBM and CBS should be defined.

13.   Section 4.2.1 deals with data “enhancement”. I believe “data augmentation” is a more accepted term, as images are not really “enhanced” in the sense of making them better, but just different to increase the amount of data, not their quality.

14.   I understood that K-means++ is used to obtain proper anchors from the industrial defects datasets, however the sentence: “The anchor frames in the *original YOLOv5* were obtained from the Microsoft public dataset training, due to the large dif- ferences between the COCO and VOC datasets and the industrial surface defect dataset” seems to say the contrary, I think this issue needs some clarification.

15.   Finally, it is stated that the PCB dataset was used for testing. But it is not clear for me if the final architecture of the proposed model was retrained with that dataset. I understand that it was, as the defect are not the same, but it should be make clear in the text. 

Author Response

(The authors gave the same response as above.)
